# Multi-Media and Multi-Band Based Adaptation Layer Techniques for Underwater Sensor Networks

**Delphin Raj K M [1]** , **Sun-Ho Yum [1]**, **Eunbi Ko [2]**, **Soo-Young Shin [3]**, **Jung-Il Namgung [4] and Soo-Hyun Park [2],***

[1]  Department of Financial Information Security, Kookmin University, Seoul 02707, Korea
[2]  Department of Computer Science, Kookmin University, Seoul 02707, Korea
[3]  Special Communication Research Center, Kookmin University, Seoul 02707, Korea
[4]  BLUnomous, Kyunggido 12285, Korea
*   Correspondence: shpark21@kookmin.ac.kr; Tel.: +82-2-910-4559

**Abstract:** In the last few decades, underwater communication systems have been widely used for the development of navy, military, business, and safety applications, etc. However, in underwater communication systems, there are several challenging issues, such as limitations in bandwidth, propagation delay, 3D topology, media access control, routing, resource utilization, and power constraints. Underwater communication systems work under severe channel conditions such as ambient noise, frequency selectivity, multi-path and Doppler shifts. In order to collect and transmit the data in effective ways, multi-media/multi-band-based adaptation layer technology is proposed in this paper. The underwater communication scenario comprises of Unmanned Underwater Vehicles (UUVs), Surface gateways, sensor nodes, etc. The transmission of data starts from sensor nodes to surface gateway in a hierarchical manner through multiple channels. In order to provide strong and reliable communication underwater, the adaptation layer uses a multi-band/multi-media approach for transferring data. Hence, in this paper, existing techniques for splitting the band such as Orthogonal Frequency-Division Multiple Access (OFDMA), Frequency-Division Multiple Access (FDMA), or Orthogonal Frequency-Division Multiplexing (OFDM) are used for splitting the frequency band, and the medium selection mechanism is proposed to carry the signal through different media such as Acoustic, Visible Light Communication (VLC), and Infrared (IR) signals in underwater. For the channel selection mechanism, two phases are involved: 1. Finding the distance of near and far nodes using Manhattan method, and 2. Medium selection and data transferring algorithm for choosing different media.

**Keywords:** multi-band; multi-media; adaptation layer; underwater communication; visible light communication (VLC); infrared (IR); acoustic

## 1. Introduction

In an underwater constrained environment, existing communication mechanisms consist of single medium and single band technology for transferring data through wireless communication. Therefore, it is difficult to apply various types of applications underwater. In existing underwater wireless communication systems, it is hard to satisfy the real time performance and reliability requirements while maintaining the connection with various heterogeneous networks beyond the application domain. In order to overcome this, a method for bundling underwater wireless media and underwater wireless bands which can adapt with the existing communication is proposed. Based on the characteristics of each medium such as acoustic, optical, IR, Magnetic Field (MFAN), etc., the adaptation layer for underwater multi-media/multi-band is proposed.

### 1.1. Needs of Multi-Band Communication Techniques Underwater

Due to the constrained conditions underwater, communication systems face problems such as multi-path, frequency fading, doppler shifts, etc. Multi-band techniques can be used as an effective technique with respect to efficiency and throughput, and to increase the performance level. Multi-band can overcome the frequency fading problem by allotting the same information to several frequency-bands.

### 1.2. Need of Multi-Media Communication Techniques Underwater

In order to get reliable communication underwater, multi-media techniques can be used. The network architecture of underwater communication includes an underwater area network, a surface area network, and a terrestrial area network. The underwater area network supports acoustic, optical, and MFAN communication; the terrestrial and surface area network consists of RF communication and the internal communication portal is TCP/UDP. In order to avoid interference and data loss, and to decrease the error rate and increase reliability, a four-in-one adaptable modem should be designed. So, different channels can be adapted in underwater communication systems. Thus, the communication system can be more effective and reliable. Also, multi-media communication systems can avoid collisions.

### 1.3. Benefits of Multi-Media/Multi-Band Communication Techniques Underwater

1.  Increase the lifetime of sensor nodes
2.  Increase the reliability of data transmission
3.  Improve the faster discovery of neighbor nodes
4.  Reduce the transmission delay between nodes
5.  Long-term connectivity between nodes
6.  Faster medium selection mechanism to transfer data

## 2. Underwater Communication Technology Overview

In this section, existing underwater technologies are described. Figure 1 depicts the architecture of the underwater environments of numerous technologies pertaining to communication. Signal communication in such environments might comprise several aspects, such as relations to buoy or ship, from terrestrial to satellite. It is also possible to exchange information using RF antennas set up on floating equipment and land stations. The exchange of data with underwater stations is made possible by using floating structures that contain communication devices. Inside water environments, it is possible to deploy various kinds of communication nodes containing AUVs, wired networks, and wireless systems in local areas. Some devices may be attached or anchored to the seafloor.

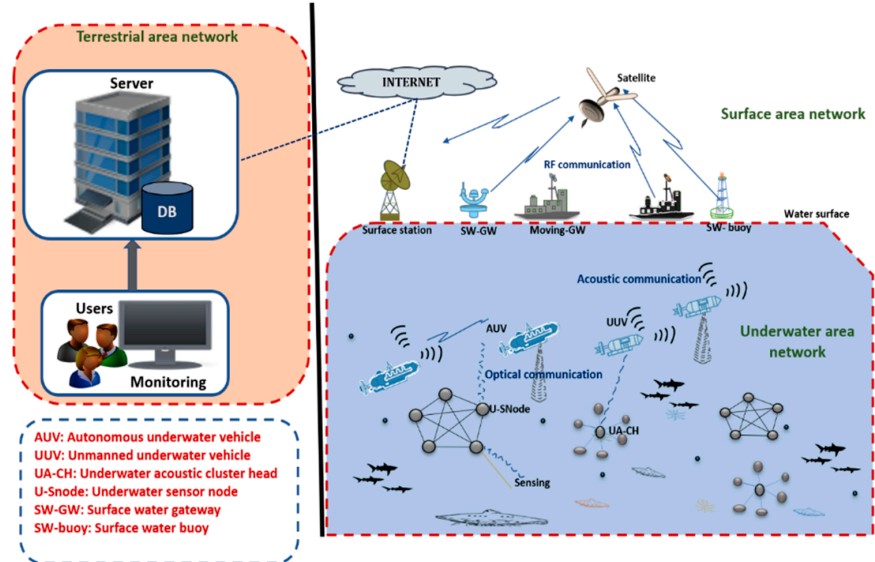

**Figure 1.** General underwater architecture and communication technologies.

### 2.1. Radio Frequency (RF) Communication

From the physics perspective, for frequency ranges of satellites, TV, mobile, and radio communications, the conductivity in seawater is very high; thus, it strongly affects EM wave propagation. Due to this, establishing communication links in ultra and very high frequency (UHF and VHF respectively) for a distance more than 10m in ocean is not very likely. In the case of lower frequencies, i.e., at very low and extreme frequency ranges (VLF and ELF, respectively), EM attenuation can be considered short enough to allow reliable communication over few kilometers to occur. Frequencies from 3 kHz to 30 kHz and 3 Hz to 3 kHz are not wide enough to allow transmissions at elevated data rates.

### 2.2. Optical Communication

The medium behavior is the main difference between optical and RF propagation in seawater. It is dielectric for optical propagation and conductor for RF. The description for this phenomenon is based upon the plasma frequency, which behaves as either a conductor or dielectric, according to the frequency range. At around 250 GHz, seawater changes from conductor to dielectric. EM waves have lower attenuation in dielectric media than in conductor ones. For a propagation range restricted to 10s of meters, higher data rates can be provided using optical technology. The effect and Doppler spread are minor in communications of optical wireless systems, as the speed of light is larger than the acoustic wave propagation speed in fluids, i.e., by around 4 to 5 orders of magnitude.

### 2.3. Acoustic Commiunication

As mentioned, RF transmission and optical transmissions have narrow propagation ranges. The first is rigorously affected by heavy attenuation, leading to a minor propagation distance, while the latter depends on the turbidity of water. So, acoustic communication is another technology which may be applied to greater distances, and is presently the leading technology for wireless communication underwater. The waveform's propagation speed depends on the medium's EM or mechanical properties. EM waves can propagate through air at a speed near to that of light in a vacuum, i.e., about 4 to 5 orders of magnitude greater than the propagation speed of acoustic waves in fluids. This levies tremendous checks on the complete transmission process by means of acoustic waves. Certainly, in acoustic-based communications, the parameters for the propagation speed play a very significant role. Underwater, the model for sound speed is calculated up to 1000 m.

## 2.4. Visible Light Communication (VLC)

The VLC is derived from optical communication. The wavelength of VLC ranges from 450 nm to 550 nm. VLC uses the blue-green spectrum. The distance for communication is up to 100 m. The speed of communication is 500 Mbps, which is very high, i.e., up to 1.5 m. VLC is perfect for one-to-one communication.

## 2.5. Magnetic Induction (MI)

This way of communication is mainly used below the sea floor. The communication distance is approximately up to 10 m. The speed of communication is $3 \times 10^8$ m/s. The data rate is in kbps.

## 3. Limitation and Advantages of Underwater Communication Technology

In this section, the advantages and disadvantages of different communication technologies underwater such as attenuation, speed, data rate, distance, etc. are described. Tables 1 and 2 show a comparison between different communication technologies.

**Table 1.** Comparison of different communication technologies in underwater.

| Parameters | Acoustic | RF | Optical |
|---|---|---|---|
| Attenuation underwater | Distance and frequency band dependent (0.1–4 dB/km) | Frequency dependent on (3.4–5 dB/m) | 0.39 dB/m (in ocean) 11 dB/m (in turbid water) |
| Speed of signal (m/s) | $\approx 1500$ ms$^{-1}$ | $2.3 \times 10^8$ ms$^{-1}$ | $2.3 \times 10^8$ ms$^{-1}$ |
| Data transfer rate | $\approx$kbps | $\approx$Mbps | $\approx$Gbps |
| Delay in communication | High | Moderate | Low |
| Approximate Distance | $\approx$km | <10.5 m | $\approx 10-99$ m |
| Bandwidth | 1–100 kHz | $\approx$MHz | $\leq 150$ MHz |
| Frequency band | 10–15 kHz | 30−300 Hz | $\approx 5 \times 1014$ Hz |
| Transmission power | >10 W | mW−W | mW−W |
| Antenna size | 0.1 m | 0.5 m | 0.1 m |
| Performance parameter | Temperature, pressure and salinity | Conductivity | Absorption and turbidity |

**Table 2.** Comparison of different communication technologies in underwater.

| Parameters | MI | Low Frequency | IR |
|---|---|---|---|
| Attenuation | 30~300 kHz | 3~30 kHZ | Depends on the distance |
| Speed(m/s) | $3 \times 10^8$ m/s | - | - |
| Data rate | ~kbps | ~Hundreds bps | ~Gbps |
| Latency | Very low | Very low | Low |
| Distance | $\approx 10$ m | Up to hundreds Kms | $\approx 3$ m |
| Bandwidth | - | - | MHz |
| Frequency band | - | - | - |
| Transmission power | $10^{-8}$ W | - | - |
| Antenna size | 0.1m | - | - |
| Performance parameter | Temperature, pressure and salinity | - | - |

## 4. Challenges Faced by the Acoustic Signal in Underwater Communication Technology

In underwater communication, the most commonly used technology is acoustic-based. The characteristics and challenges of acoustic communication that affect current systems are shown in the Section 4.1.

### 4.1. Challenges in Acoustic Signal

1. Low bandwidth: In underwater environments, the noise is very high because of very low medium frequencies. The bandwidth is highly dependable on the distance of the transmission, that can be lesser than 1 KHz [1].
2. High channel error rate: The interference in underwater acoustic channels is due to fading and multipath. This can cause connection losses due to high channel error rates in acoustic communication [1].
3. Large propagation delay: The speed of underwater acoustic communication varies from 1480 m/s to 1540 m/s [2,3]. Hence, the propagation delay is five times higher than the radio frequency (RF) in terrestrial area networks. Similarly, there is an extreme level of variation in propagation delay depending on the water pressure, temperature, salinity, etc. Though propagation delay is the critical point for underwater acoustic sensor networks, deep designs for MAC protocols are considered [4].
4. High energy consumption: In acoustic communication, the magnitude is higher when compared to the terrestrial environment. The ratio of power usage is even higher when compared to the terrestrial environment. The batteries in the constrained environment cannot be rechargeable [5].
5. Low memory storage: The memory storage level of the underwater nodes is less than that of terrestrial nodes [6].
6. Problems with sensors due to fouling: The sensor nodes that are deployed in underwater environments can accumulate waste materials such as soil, fish waste, oils, etc. This can affect the operation of sensor nodes [6].
7. Multipath and surface scattering: Due to multipath and surface scattering, when the signal is sent from the source to destination via different paths, the strength of the signal could decrease. This causes problems in data transmission [7].
8. Problems in routing: Routing in underwater communication faces challenges such as reliability in data transmission, network connectivity, forwarding of data, and variations in the link. Therefore, in underwater communication, routing is the most difficult issue. Hence, a multi-channel/multi-band adaptation layer is needed in underwater communication.

### 4.2. Challenges in MAC Protocol

In Underwater acoustic sensor networks, medium access control is still an unsolvable problem. The challenges faced by the MAC protocols are listed below.

1. Topology design: The underwater network's infrastructure always changes due to node mobility, node failure, adding new nodes, etc. The performance of MAC protocol depends on the topology design. So, topology design is a critical task in the underwater environment.
2. Sensor node deployment: In underwater networks, the sensor nodes are sparsely deployed in different places. Long-range communication depends on the availability of connections between the nodes. So, in the MAC protocol, the design of the node deployment is a critical issue.
3. Time synchronization: In the MAC protocol, the power cycling method works on time synchronization. In order to handle time certainty between nodes, time synchronization is necessary in MAC protocols.
4. Power wastage: The power wastage in sensor nodes is because of collisions while transmitting data. So, MAC protocols must be designed to avoid collisions between nodes.

5.    Other Issues:  Also, the MAC protocol underwater gives rise to other problems, such as making connections with centralized networks, high delays for handshaking, collision avoidance problems, etc.

## 5. Analyzing the Challenges and Advantages of Protocols in Underwater Communication

### *5.1. MAC Layer Protocols*

In this section, the MAC protocols designed and developed for underwater communication are described, and their advantages and disadvantages are noted.

### 5.1.1. Contension-Free Based MAC Protocol Design

In 2009, a CDMA-B was developed whose main purpose was to save energy. Near-far signal is one of the major problems affecting performance levels underwater [8]. In 2011, POCA-CDMAMAC was developed, where the round-robin methods in it are used to receive packets from neighboring nodes. All nodes sending the packet in the same interval of time are the disadvantage [9]. In [10], the PLAN-MAC protocol was developed for long latency access networks. In this approach the contention free algorithm is used to spread the codes, i.e., each node will get a unique code. The staggered TDMA Underwater MAC Protocol (STUMP) [11] was developed in 2009, where there is no need of synchronization between nodes in order to avoid overlap during transmissions. The disadvantages of STUMP are the usage of a number of time slots. ER-MAC [12] was developed in 2008. This is an energy efficient protocol, but it is not suitable for multi-hop communications. WA-TDMA [13] was developed in 2009; the sleep-awake method is used for each node to avoid energy wastage. In WA-TDMA, slot allocation is one of the major issues. ACMENet [14] was developed in 2009; the sleep-awake method is used for each node to avoid energy wastage. Slot allocation is the main issue in this protocol. Dynamic Slot Scheduling Strategy (DSSS) [15] was developed in 2011 for the best usage of channels in underwater communication. The transmission pairs are increased to avoid collisions and parallel data transmissions. Synchronization is the essential requirement in this approach. UW-FLASHR [16] was developed in 2008; tight synchronization is not required in its use. There is a time gap during the transmission of data. ST-MAC [17] was developed in 2009; it describes the delays between the transmission links openly and modelled the ST-MAC as a new technique for solving vertex coloring problem. OFDMA [18] was developed in 2009; in it, topology is centralized. This protocol is used to solve the hidden terminal problem, but many users can affect the performance of the system. UW-OFDMAFC [19] was developed in 2009; in it, the topology is centralized. This protocol is used to solve the hidden terminal problem. Many users can affect the performance of the system.

### 5.1.2. Contension Based MAC Protocol Design

S-Aloha [20] was developed in 2006. In underwater acoustic communication, the propagation delay is high, so there is no coordination between nodes. Based on an analysis, the system performance is not much different from the Aloha and S-Aloha systems. T-Lohi [21] was developed in 2007; this mechanism exploits space-time uncertainty problems and high latency problems to detect collisions. The power consumption is low for wake-up, and this protocol gives good throughput. CUMAC [22] was developed in 2012; it is considered a cost-effective technique, thanks to its use of only one transmitter to transmit the data. The hidden terminal problem is still not solved in this approach. MACA-MN [23] was developed in 2008. This approach solves the hidden terminal problem. It can increase the packet delivery rate compared to MACA. R-MAC [24] was developed in 2007; in it, data and control packets are scheduled on both sides such as sender and receiver. UMIMO-MAC [25] was developed in 2011. UMIMO-MAC is designed to: (1) adaptively leverage the tradeoff between multiplexing and diversity gain, (2) select suitable transmit power to reduce energy consumption, and (3) efficiently exploit the UW channel and minimize the propagation delay underwater. Channel Stealing MAC (CS-MAC) was developed in 2011 [26] to solve the hidden and exposed terminal problems. This protocol is

satisfactory regarding channel utilization and delay in transmission. MACA adaptation for underwater network (MACA-U) [27] was developed in 2008. In this approach the MACA protocol was tested for underwater communication, focusing on areas such as transaction rules, forwarding packets, and back-off methods. The performance evaluation was good in multi-hop underwater communication. Contention-based Parallel rEservation MAC (COPE-MAC) [28] was developed in 2010 for underwater acoustic communication. In this approach, two techniques, parallel reservation and cyber carrier sensing, are introduced. The advantage of this protocol is its ability to increase the throughput; furthermore, it is very comfortable in large networks. Multiple-rendezvous Multichannel MAC (MM-MAC) [29] was developed in 2010. In this mechanism, the cyclic quorum approach was used to reduce the probability of collisions in underwater communication; it increased the performance in multi-hop underwater wireless sensor networks. The Receiver Initiated Packet Train (RIPT) protocol was developed in 2008 [30]. This protocol was developed to address propagation delays in underwater channels. Its advantage over other protocols is the throughput level, because of the minimum collision rate.

### 5.1.3. Hybrid MAC Protocol Design

Hybrid Spatial Reuse (HSR-TDMA) was proposed in 2010. This technique has been applied in underwater ad-hoc networks which use the spatial reuse methods to improve the throughput level. The experimental result showed that the availability of underwater nodes increased to transmit the data [31]. Hybrid Medium Access Control Protocol (H-MAC) was developed in 2010. This protocol was proposed to improve the power efficiency in traffic, quality of service, channel utilization, etc. [32]. Pattern-MAC (P-MAC) was developed in 2005. The sleep-wake schedule of underwater nodes was adaptively determined based on the traffic condition and the neighbor nodes [33]. UW-MAC, also known as CDMA-based energy control MAC, was proposed in 2010. This protocol focuses on delay, network throughput, and increased network lifetime [34].

### 5.2. Routing Layer Protocols

In this section, the routing protocols designed and developed for underwater communication are described and the advantages and disadvantages of routing protocols are noted.

### 5.2.1. Routing Protocols Including Localization Techniques

SEANAR [35] was developed in 2010; it acts as a power efficient routing protocol for underwater communication. SEANAR shares topology information along with other information with the neighboring node. REBAR [36] was developed in 2008. In this approach, a cylindrical path is created between source and destination. This method is used for energy balancing in underwater routing techniques. Its major disadvantage is the difficulty of finding the position of the node due to node mobility. A Depth Adaptive Routing Protocol DARP [37] was developed in 2012; this approach is based on the depth of water and the speed of signal changes. The test proves that when the depth is under 1000 m, the signal strength is good, and communication is faster. The result shows that it will reduce the delay between end-to-end communication. Lifetime-Extended Vector-Based Forwarding (LE-VBF) [38] was developed in 2012. This approach is used for energy saving. The Mobicast [39] routing protocol, also known as mobile geocast, was developed in 2012. The data collection efficiency is solved by using this protocol. This approach comprises two phases: in the first phase, it collects the data inside the 3-D ZOR; in the second, the nodes in the 3-D ZOR are woken up. Because of this approach, power can be saved by activating the nodes only inside the 3-D ZOR. HH-VBF [40] was developed in 2008. This protocol is used to reduce the data load while routing through VBF. Computational complexity is the major issue associated with this approach. Vector-Based Void Avoidance (VBVA) [41] was developed in 2009. The vector shift routing strategy is used in this approach. One of its advantages is a high packet delivery rate; the major disadvantage is its high- power consumption and delays in data delivery.

### 5.2.2. Routing Protocols without Localization Techniques

DBR [42] was developed in 2008. In it, depth-based sensor nodes are used to send the data from bottom to top. The throughput rate is high for this approach. Its disadvantage is high power consumption. Q-ERP [43] was developed in 2017. The major advantages of this protocol are the high packet delivery rate, power efficiency, and absence of delays in data delivery. Adaptive Mobility of Courier nodes in the Threshold-optimized DBR Protocol (AMCTD) [44] was developed in 2013. The main advantage of this protocol is its improvements in the lifetime of the underwater network. The information-carrying based routing protocol (ICRP) [45] was developed in 2007. The main advantages of this protocol are its scalability and power efficiency. Low data delivery rates among its major disadvantages. The multi-layered routing protocol (MRP) [46] was developed in 2014. In this protocol, two phases are used: 1. a layering phase: In this phase the layers are formed towards the super node; and 2. A data forwarding phase: The data is forwarded through this formed layer.

### 5.3. Transport Layer Protocol

In this section, the transport layer protocols designed and developed for underwater communication are described and their advantages and disadvantages are noted.

Multi path and network coding (MPNC) was developed in 2015. It is considered a reliable protocol for underwater acoustic communication [47]. Twin path and network coding (TPNC) were developed in 2015. The ratio of data delivery is the same as that of MPNC, and power consumption is low in comparison [47]. Th erasure code based multi-hop reliable data transfer scheme (ECRDT) was developed in 2017 for underwater wireless sensor networks. This approach uses a packet level forward error correction method in end to end codes [48]. The adaptive RTT-driven transport layer flow and error control protocol (ARTFEC) was developed in 2014. Data flow control techniques are used here to apply various characteristics to the acoustic channel. In this approach, the reliability and data transmission rate are high [49]. Segmented data reliable transfer (SDRT) was developed in 2010. This protocol transfers packets in blocks, and it improves the utilization of underwater channels. Power consumption is also moderate in this approach [50].

## 6. Related Studies Based on Multi-Band and Multi-Media Communication Technologies

### 6.1. Multi-Band Underwater Communication

A multiband OFDM for converting sound signal is used here. For low SNR communication, a transmitter and receiver are used. Multi-band OFDM techniques can reduce the complexity on the receiver side compared to the single-band OFDM techniques. The proposed scheme is tested in the sea, covering a total bandwidth of 3.6 kHz, with 16 sub-band systems, a data rate of 4.2 and 78 bits/s, and a range of 52 km. The limitation of OFDM signals was clearly revealed in 78 bit/s, but the performance level was low because of a failure in the signal and synchronization problems at 4.2 bit/s [51]. The performance degradation of one band affects all the other bands; thus, the performance level of multi-bands is worse in comparison. In order to solve this problem, the error rate of each band should be analyzed on the receiving side, and then the threshold should be set, and lesser weights allocated to the inferior bands. An algorithm is used to set the threshold for preamble error rates. In this experiment, the performance level was increased as the number of multi bands increased [52]. The dynamic bandwidth schedule algorithm was proposed to attain a multi-condition bandwidth in optical networks using OFDM-PON [53]. For a 21-inch autonomous underwater vehicle used by the navy, acoustic communication with multiple data rates and two frequency bands was developed. This system includes high and mid frequencies modems, i.e., 25 kHz and 3 kHz, and data rates of 80 bps to 5000 bps respectively, to increase the reliability of the system [54]. The normalized match filter approach is proposed based on the frequency domain processing method. The performance of diver's signal was evaluated in the Hudson river. The frequency bands with highest SNRs were used [55].

## 6.2. Multi-Media Underwater Communication

[56] For underwater acoustic communication, MC-UWMAC was designed. This protocol is developed based on low power and multiple channels. Collision-free communication is guaranteed. [57] The MAC protocol for underwater acoustic communication was proposed to solve the presence of noise sources around the region covered by network. Noise-aware MAC (NA-MAC) is a protocol that improves the ability of nodes; it was developed with a multi-band modem using the frequency band to detect increases of in-band noise.

NA-MAC working flow [57]:

1. START: In the starting stage, the default-band can be used by the nodes.
2. ALERT: In this stage, the nodes should be aware of increases in noise, so that they might change to a new frequency band.
3. TRACKING: In this stage, the nodes exchange their information with neighboring nodes using PREQ (packet used for noise level request) and PRES (packet used to replay to noise level request) to get the updates about the noise level.
4. NEW-CHANNEL: The node changes the current band and transmits through a new band.

## 6.3. Multi-Band Techniques for Adaptation Layer

In this multi-band technique, existing methods, such as OFDM, FDMA, or OFDMA, can be used to divide the frequency in underwater communication. In [58], OFDM has many advantages for underwater communication schemes. Limited underwater acoustic bandwidth can be utilized efficiently by the use of OFDM. OFDM makes effective use of the spectrum by allowing overlay between sub-carriers. The introduction of guard time with cyclic prefixes reduces inter symbol and inter carrier noise significantly; hence, the modulation scheme is robust against ISI and ICI. By dividing the wide band frequency selective channel into narrowband smooth fading sub channels, OFDM is more robust in terms of frequency fading. It is computationally-resourceful thanks to the IFFT and FFT methods which implement the modulation and demodulation functions respectively. In addition to the advantages of the basic OFDM scheme, the performance of the communication scheme for underwater channels can be further improved in following ways. By the introduction of appropriate pilot carriers, effects due to channel distortion can be corrected. The introduction of Forward Error Correction (FEC) coding and interleaving can expand the reliability and performance of the communication scheme by significantly reducing the Bit Error Rate (BER).

The OFDMA-based MAC protocol is constructed on the OFDMA technology, which splits an accessible channel into a several orthogonal sub-channels, called "subcarriers". We use this technology to allow concurrent sessions through subcarrier sharing among the nodes that are in communication with each other. Each nearby pair of nodes uses a subcarrier or a pair of subcarriers to send data. This used set of subcarriers is, therefore, kept for the pair until they relinquish it clearly. The timeline is separated into slots, each of length Ts, where transmissions start only at the beginning of each time slot. Synchronization is assumed to be done through a one-hop transmission from the base station. The objective is to achieve optimal sharing of the available subcarriers; optimal here means the best distribution of the available subcarriers among network nodes which results in the minimum transmission power consumption subject to a minimum required throughput level [58]. Figure 2 shows the scheme for OFDM-based underwater communication. This method can be applied to our multi-band techniques for splitting the frequency band.

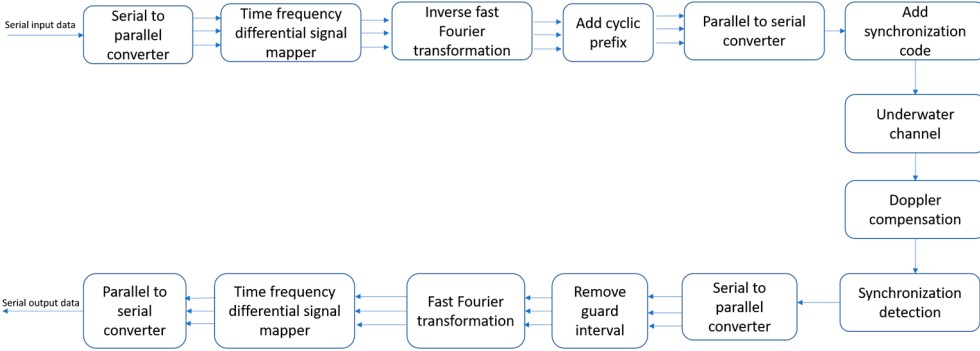

**Figure 2.** Communication scheme of the OFDM approach.

The parameters used for OFDM-based communication are bandwidth, carrier spacing, carrier frequency, sampling frequency, cyclic prefix duration, symbol duration, total symbol duration, subcarriers, etc.

## 7. Proposed Scheme

### 7.1. Protocol Stack of Mult-Band and Mult-Medium Techniques

Existing underwater communication consists of single medium and single band technology for transferring data through wireless communication. It is difficult to apply various types of applications underwater. Figure 3 shows a protocol stack of multi-band and multi-media underwater communication. The protocol stack consists of different layers, i.e., the physical, data-link, network, transport, and application layers. The adaptation layer is an extension of the data-link layer which contains multi-band and multi-media technology. The multi-band approach is used to split the bandwidth into different frequencies, while the multi-media approach is used to share the data through different channels such as acoustic, visual light, infrared, etc. Multi-band and multi-media approaches are used for the reliable transmission of data in underwater communication.

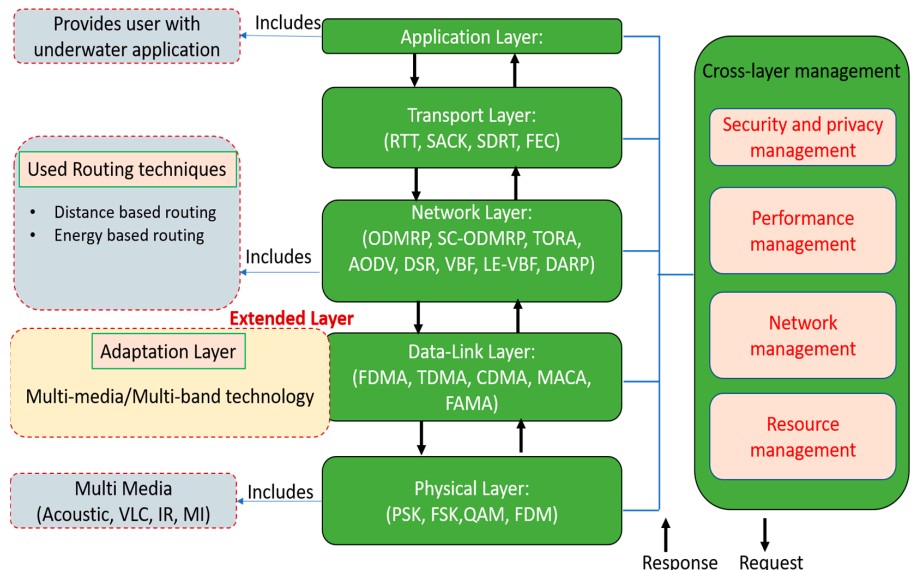

**Figure 3.** Protocol stack design for multi-band/multi-media techniques.

### 7.2. Proposed Adapation Layer Scheme

Figure 4 shows the adaptation layer mechanism. In this case, the MAC layer is formed as the extended layer to create the adaptation layer. Again, the adaptation layer is divided into two techniques,

known as multi-band and multi-media techniques. A description of multi-band/multi-media techniques is given below.

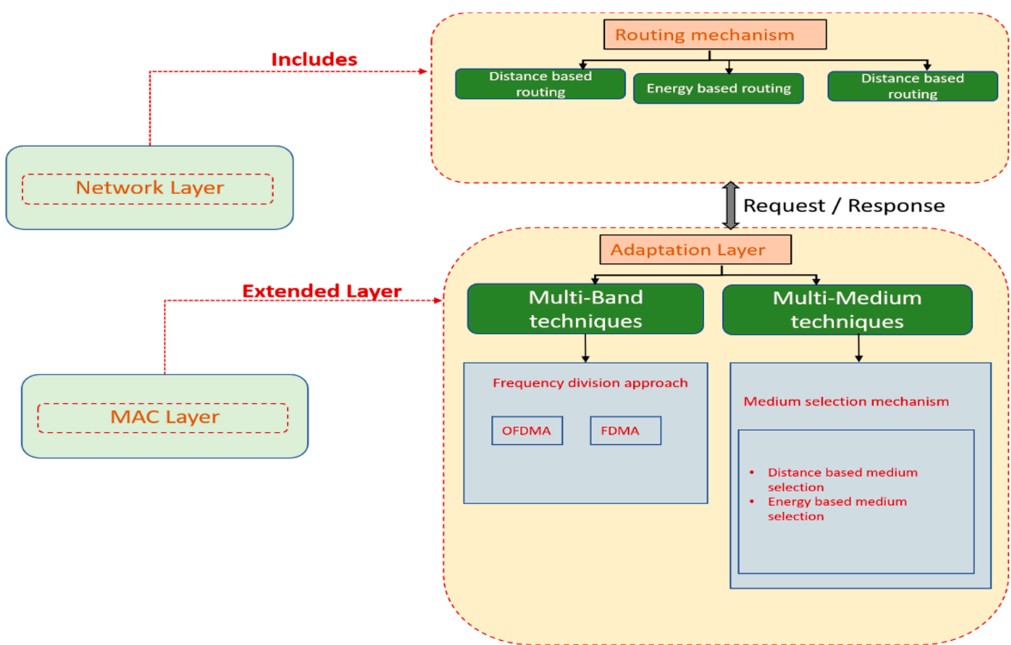

**Figure 4.** MAC extension and adaptation layer.

*7.3. Modem Design of Proposed Scheme*

In this section, the components used for designing the adaptation layer are shown. The basic components of underwater multi-media/multi-band communications links are shown in Figure 5. The main component of the modem hardware is the transmitter, which contains components such as a modem controller, medium switch controller, power controller, modulator, frequency band splitter, etc. The medium switch controller (MSC) controls the switch to select the type of channel, e.g., VLC, IR or Ultrasonic. Then, the modulator contains the desired information in light or acoustic form. The acoustic Tx is the acoustic transmitter used to transmit the acoustic signal through the underwater channel. The VLC Tx and IR Tx are used to transmit an optical signal through an underwater channel. At the receiver end, the detector is used to receive the acoustic or optical signal. The received signal process demodulates to get the original data in the form of bits.

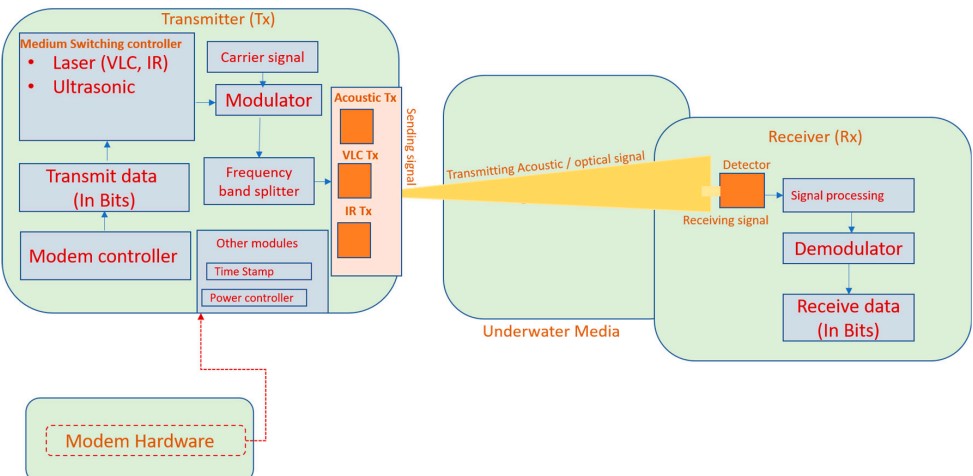

**Figure 5.** Physical layer components and workflow.

### 7.4. Medium Selection Mechanism

In this section, multi-media techniques in the adaptation layer will be discussed. Multi-media techniques are used to select the appropriate medium from the various media in underwater communications, such as acoustic, VLC, or IR. Figure 6 shows a means of communication scheme from the physical to the routing layer, and the placement of the medium selection mechanism inside the MAC layer.

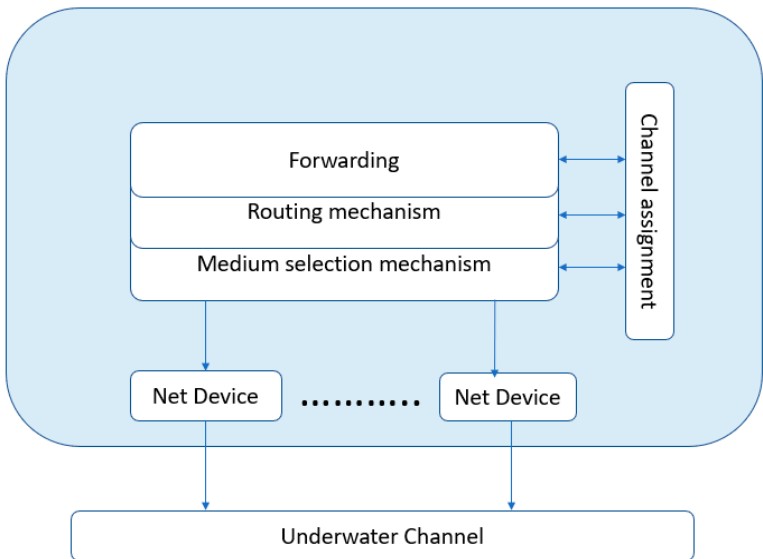

**Figure 6.** Medium selection design scheme inside adaptation layer.

Multi-media techniques consist of two phases 1. Distance calculation 2. Medium selection and data transfer

### 7.4.1. Distance Calculation

This phase is used to find the distance between nearby and distant nodes in underwater communication. For distance finding methods, we need to follow two steps: a.) The received signal strength indicator (RSSI) method is used to find the signal strength between the nodes, and b.) the Manhattan method is used to find the distance between nearby and faraway nodes.

Received Signal Strength Calculation

RSSI is used to calculate the loss of signal between the nodes in underwater sensor networks, including signal loss in underwater networks. The propagation model is used to calculate the RSSI between the nodes. The formula with which to measure the path loss is shown below

$$Path\ Loss(d) = Path\ Loss(d0) + 10nlog\left(\frac{d}{d0}\right) + X\sigma \qquad (1)$$

In Equation (1), d is the distance between the nodes. Based on the decrease in RSSI, the path loss n should be measured. The value of $X\sigma$ is 0, which is the Gaussian distribution variable to change the power of the received signal at a certain distance. The referential distance is d0, which is equal to 1 m. Path Loss(d0) is the referential power value.

Suppose R to be the received signal strength at distance d0 between the transmitter and receiver nodes; then, the equation can be written as follows

$$R(dB) = Pr - Path\ Loss(d0) \qquad (2)$$

In Equation (2), Pr is the power of node that is transmitting the signal and Path Loss(d0) is the referential power in dB. The RSSI value can be calculated using the formula shown in Equation (3).

$$RSSI(dB) = R - 10n\log d \tag{3}$$

Distance Estimation of Nearby Node Using Manhattan Method

In [59,60], the Manhattan method was used to find the distance between near-far nodes. In this case, the referential object must be allocated. Let's consider the referential set as $Ref_{ab}$ for the pair of nodes (a, b), where a is the nearby node of b. Let $N_a$ and $N_b$ denote the nearby set of nodes a and b. Then, the referential set can be created as $Ref_{ab} = N_a \cup N_b \cup \{a\} \cup \{b\}$.

Based on the referential set $Ref_{ab}$, two vectors $V_a$ and $V_b$ can be generated for nodes a and b respectively by using vector generation. In this case both comprise a RSSI value in dB for all the nodes in $Ref_{ab}$. The sending power of the RSSI node itself is set at 0. That is RSSI (a, a) = 0 in vector $V_a$. If the node d is far from the node a, then RSSI (a, d) is set at −100.

If the given vectors are $(a_1, a_2, a_3 \dots)$ and $(b_1, b_2, b_3 \dots)$ with elements at same number n, then the Manhattan distance $M_{dis}$ can be calculated using Equation (4).

$$M_{dis} = \sum_{i=1}^{n} ai - bi \tag{4}$$

Figure 7 shows an example of a graph constructed using the Manhattan distance. Suppose the referential set $Ref_{ab}$ of $N_{S1}$ and $N_{S2}$ is obtained using (S1, S2, S3, S4, S5). $V_{S1}$ and $V_{S2}$ are the set as [0, −55, −65, −60, −100] and [−60, 0, −50, −75, −70], which is the RSSI value. Then, the distance between the nearby nodes S1 and S2 can be calculated as shown in Equation (5).

$$M_{dis} = |0 + 60| + |-55 + 0| + |-65 + 50| + |-60 + 75| + |-100 + 70| = 175 \tag{5}$$

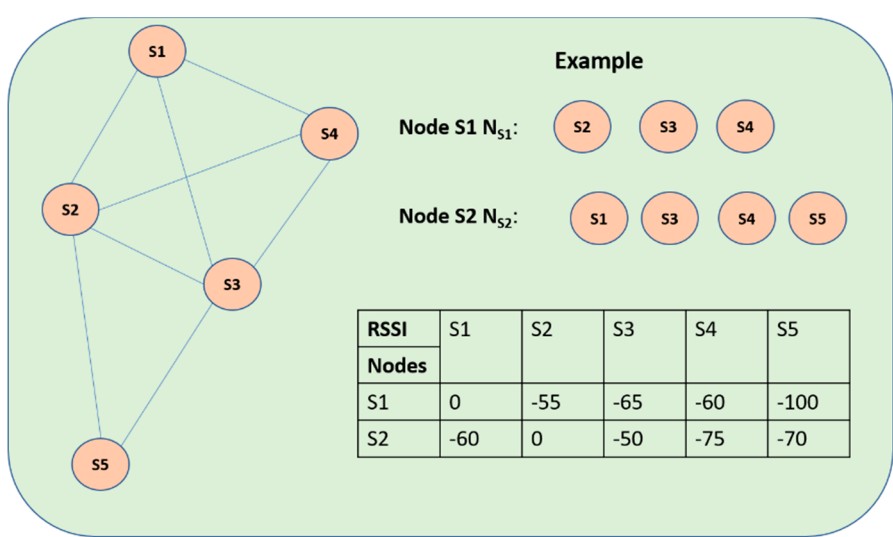

**Figure 7.** Nearby nodes distance calculation using Manhattan method.

Distance Estimation of Far-Away Node Using Manhattan Method

By using the RSSI values, the distance between the nearby nodes can be calculated. We know that the Manhattan distance of $M_{dis}$ (S1, S2) = 175, so the distance between (S2, S1) = 175. Similarly, the other distances, such as $M_{dis}$ (S2, S4), $M_{dis}$ (S1, S3), $M_{dis}$ (S2, S5), $M_{dis}$ (S1, S4), $M_{dis}$ (S3, S4), $M_{dis}$ (S2, S3), and $M_{dis}$ (S3, S5), can be calculated. Figure 8 shows the methods for finding the distance for far-away nodes. The shortest path method is applied here after finding the $M_{dis}$ of nearby nodes.

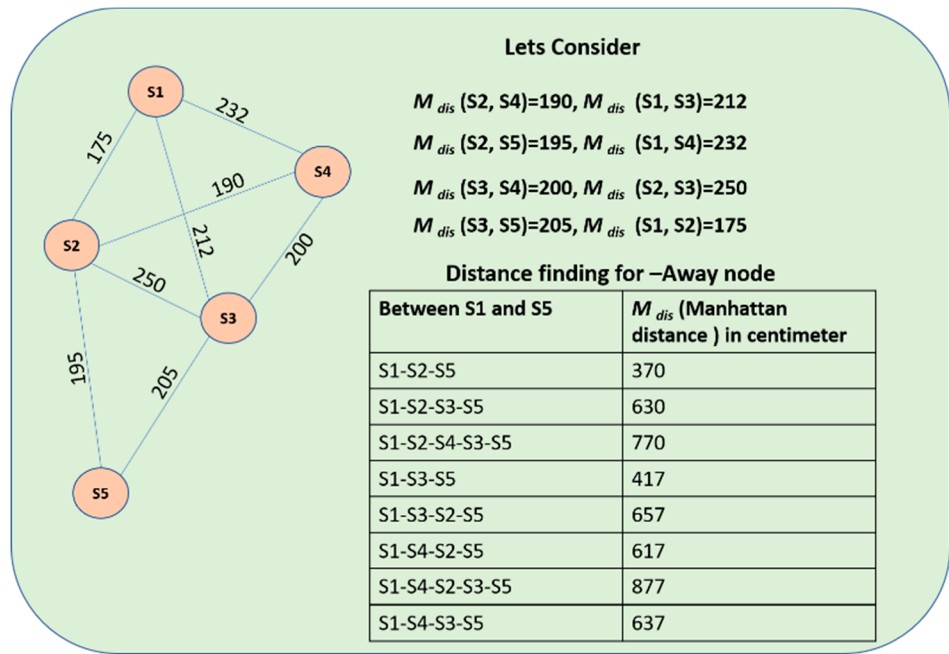

**Figure 8.** Far-away node distance calculation using Manhattan method.

### 7.4.2. Medium Selection and Data Transfer

#### 7.4.2.1. Flowchart for Medium Selection Mechanism

Figure 9 shows a flowchart of the medium selection mechanism. At first, the nodes will obtain the RSSI values and calculate the distance between the nearby nodes. If data = 1 means, there is an availability of data to send, while if data = 0, there is no data to send. If data is available, then the adaptable medium to select is needed. Here, the medium selection is based on the distance between the neighboring nodes. If distance is less than 5, the IR medium is used to send the data. If the distance is between 5 to 20 means, the VLC medium is used to send the data, while if the distance is greater than 20 means, the acoustic medium is used. The pseudo code of medium selection mechanism is given in Algorithm 1.

#### 7.4.2.2. Algorithm for Medium Selection Mechanism

---
**Algorithm 1: Medium Selection Algorithm**

---
1. begin
   a. for i = 1 to n, where n is the number of nodes
      i. Obtain the value of RSSI and calculate the distance between the nearby nodes
      ii. if data is available (data = 1)
         1. if distance is less than 5 m
               Select IR as the medium, goto step 4
         2. else if distance is between 5 and 20 m
               Select VLC as the medium, goto step 4
         3. else if distance is greater than 20 m
               Select acoustic as the medium, goto step 4
         4. Send data through the medium selected.
      iii. else if data is not available (data = 0)
         1. Wait for new data to arrive
   b. end for
2. end

---

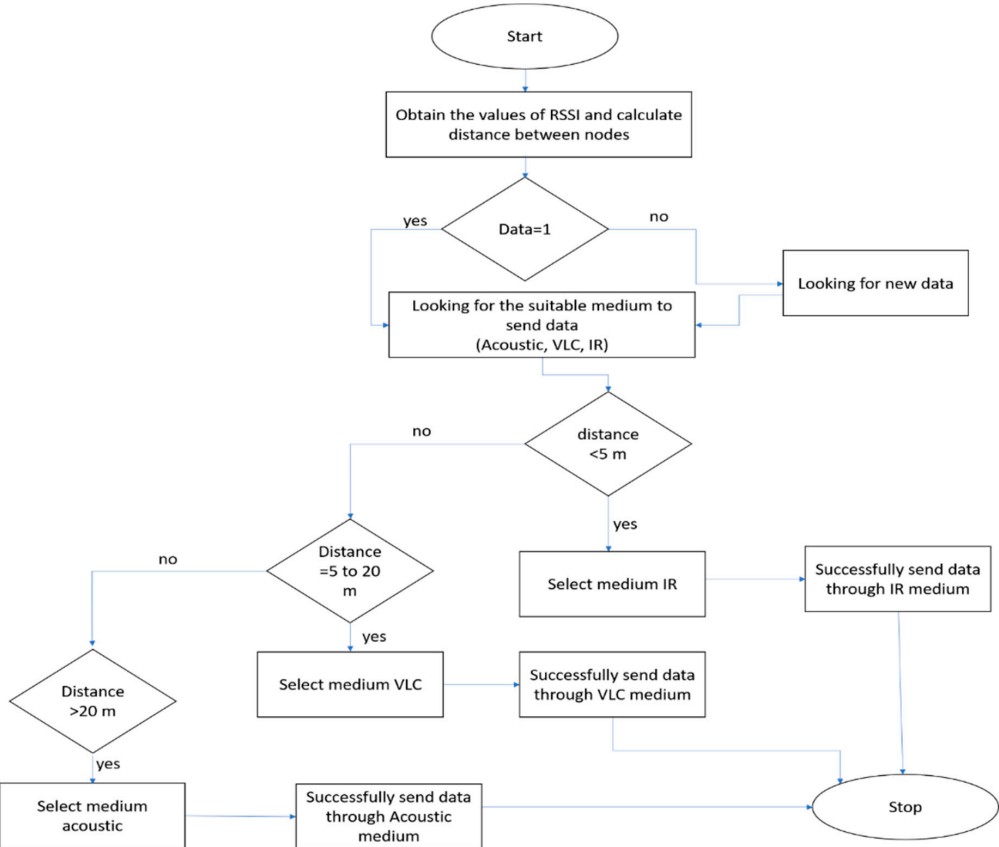

**Figure 9.** Medium selection mechanism.

### 7.4.2.3. Scenario for Medium Selection Mechanism

In Figure 10, two scenarios in which the multi-media approach can be used are presented: (1) Tsunami monitoring, and (2) Diver networks monitoring. In both scenarios, the data needs to reach the control room much faster, and that data should be reliable. So, the distance is used as the criteria for selecting the medium. The distance between the nearby nodes is calculated using RSSI values and the Manhattan method. Table 3 shows the RSSI value received as dB from the nearby node. For example, S1 receives −33 dB from node S4, which is one of the nearby nodes of S1. So, its distance is 4 m, based on the Manhattan method.

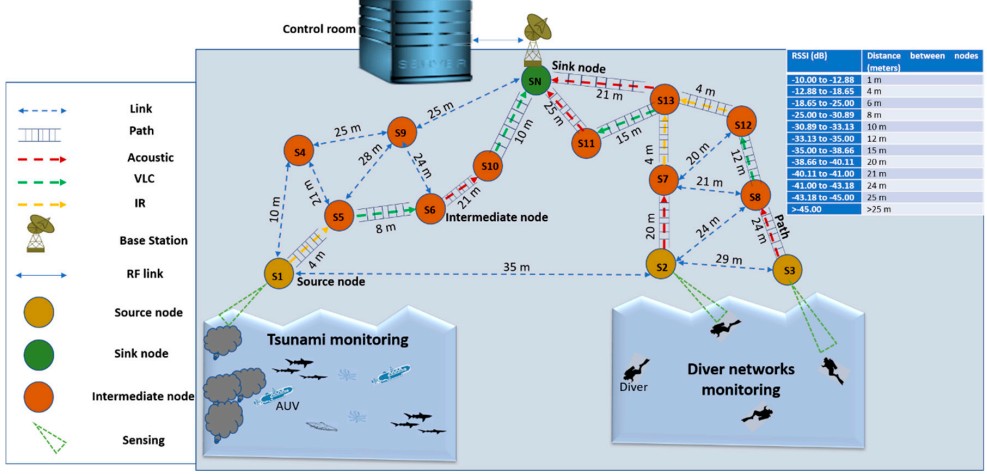

**Figure 10.** Surveillance using the multi-media mechanism.

**Table 3.** RSSI values of nearby nodes based on distance.

| RSS/Sensor Nodes | S1 | S2 | S3 | S4 | S5 | S6 | S7 | S8 | S9 | S10 | S11 | S12 | S13 | SN |
|---|---|---|---|---|---|---|---|---|---|---|---|---|---|---|
| S1 | 0 | −55.0 | - | −33.1 | −17.66 | - | - | - | - | - | - | - | - | - |
| S2 | −55 | 0 | −46.3 | - | - | - | −37.1 | −42.1 | - | - | - | - | - | - |
| S3 | - | −46.3 | 0 | - | - | - | - | −43 | - | - | - | - | - | - |
| S4 | −32.0 | - | - | 0 | −41 | - | - | - | −44.9 | - | - | - | - | - |
| S5 | −77.6 | - | - | −41 | 0 | −31 | - | - | −55.0 | - | - | - | - | - |
| S6 | - | - | - | - | −30.1 | 0 | - | - | −43 | −40.11 | - | - | - | - |
| S7 | - | −39 | - | - | - | - | 0 | - | - | - | - | −40.01 | −18 | - |
| S8 | - | −42 | −43 | - | - | - | −40.11 | 0 | - | - | - | −33.66 | - | - |
| S9 | - | - | - | - | - | −43 | - | - | 0 | - | - | - | - | −45 |
| S10 | - | - | - | - | - | −40.1 | - | - | - | 0 | - | - | - | −30.99 |
| S11 | - | - | - | - | - | - | - | - | - | - | 0 | - | −37.01 | −45 |
| S12 | - | - | - | - | - | - | −40.01 | −33.88 | - | - | - | 0 | −17 | - |
| S13 | - | - | - | - | - | - | −13.00 | - | - | - | −37.01 | −17 | 0 | −41 |
| SN | - | - | - | - | - | - | - | - | −45 | −30.9 | −45 | - | - | 0 |

Figure 11 shows the multi-media method which is applicable for reliable and fast routing underwater. The result is based on source to destination routing using the Manhattan method. In this approach, multiple media can be used for communications between nodes. In our working scenario, there are a number of sensor nodes from S1 to S13, and SN is considered as the sink node. Here, S1, S2, and S3 are the source nodes that are to transmit the data. In this case, it will choose the most efficient method for transmitting data. For example, S1 will transmit data through S1→S1-S5-S6-S10-SN using the Manhattan approach. Also, this approach will be used in different media for communication between sensor nodes based on Sections 7.4.2.1 and 7.4.2.2.

**Medium selection based on distance**

**Figure 11.** Source to sink node routing using the medium selection mechanism.

## 8. Implementation Setup and Results

### 8.1. Multi-Media Modem

Figure 12 shows the modem design and tests done using the multi-media approach. In Figure 11, 'a' represents the setup of modem and 'b' represents the operation performed using multi-media communication. The conceptual modem was developed using a combination of Raspberry Pi 3+ and Beagle-bone Black. The conceptual model setup was done with PC-to-PC link using acoustic and visible light communication. In this approach, an acoustic modem developed by [61] and the visible light (VL) scheme developed by [62] were combined. The experimental result shows that this modem can send 30 bytes of data at 1000 ms, and that it can send 1000 times each to acoustic and VL. Table 4 shows the conceptual modem's specifications.

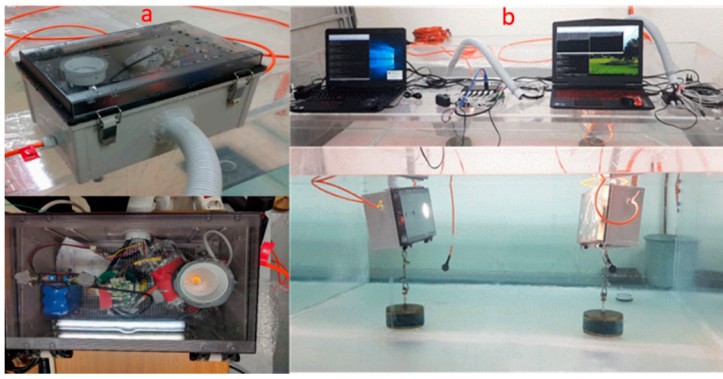

**Figure 12.** Multi-media modem setup and operation.

**Table 4.** Multi-media modem specifications.

| Specification Used | Acoustic | Visible Light |
|---|---|---|
| MCU | ATmega | ATmega |
| Protocol/Interface | Serial | Serial |
| Modulation | BPSK | OOK |
| Operational frequency | 30 KHz | - |
| Communication range | 50 m | 1.5 m |
| Power consumption | 15V, 4.5 Watt | 12V, 2.4 Watt |
| Baud rate | - | 38,400 |
| Dimension | 70 mm × 40 mm | 44 mm × 18 mm |

### 8.2. Multi-Media/Multi-Band Modem Setup and Testing

The 'a' part of Figure 13 shows the modem setup for the multi-media/multi-band approach; the 'b' part of Figure 12 shows the operation selection and working of the multi-media/multi-band device. The modem setup was developed using a Xilinx zynq board with multi-media/multi-band techniques embedded inside it. This modem is applicable for real time applications such as diver network monitoring, tsunami monitoring, etc. as shown in Section 7.4.2.2. In this approach, the modem was developed with: (1) Two transducers supporting bandwidths of 70 kHZ and 140 kHZ, as used for acoustic communication; (2) a single bandwidth for infrared (IR) communication at a wavelength of 700 nm to 1 mm; and (3) Visible light (VL) communication using the blue wavelength at a range of 450 to 485 nm. So, this modem setup is combines of multiple bands for different media, such as acoustic, VLC, and IR. The detail specifications of the modem used for the multi-band/multi-media approach is shown in Table 5.

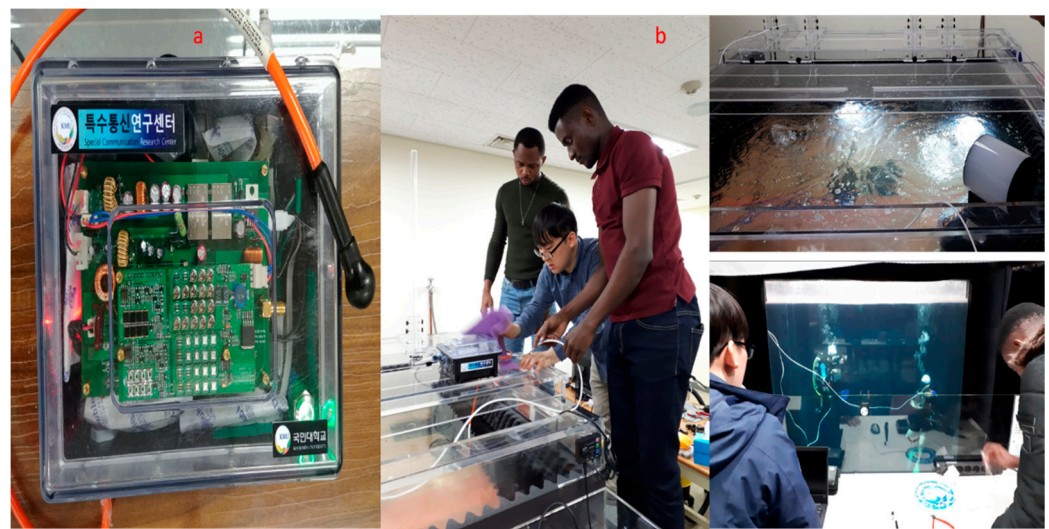

**Figure 13.** Multi-media and multi-band modem setup and operations.

**Table 5.** Multi-media/multi-band modem specification.

| Specification | Description of Modem |
|---|---|
| XILINX™ ZYNQ APSoC XC7Z020-CLG484-1 | Dual-core ARM Cortex-A9 MPCore™ with CoreSight, the operation is up 667MHz, NEON™ & Single/Double Precision Floating Point for each processor and 85K Programmable Logic Cells |
| Memory | 512MB DDR3, 256Mb Quad-SPI Flash, Full size SD/MMC card cage and 16GB SD Card (UHS-1) |
| Connection type | 10/100/1000 Ethernet, USB OTG (Device/Host/OTG), USB UART |
| Expansion | 2.54 pitch Box Header, Pmod™ headers (2 × 6) |

| Specification | Description of Modem |
|---|---|
| Indicator | User (LEDs-9) |
| Input Toggle Button | Push button switches 7 |
| Analog-to-digital | Xilinx XADC header (4 AD Input)/AD9467 |
| Transducers | [Ultrasonic Transducer] Omni-directional, Its Frequency range: 70/140 kHz [Infrared Diode] TX: 850(±20 nm) nm, RX: 750~1100 nm, Deg: ±60° [Visual Light Diode] TX: 450~460 nm, RX: 430~610 nm, Deg: ±65° |
| Debug/Programming | On-Board USB JTAG programming port and ARM Debug Access Port (DAP) |
| Electrical power source | Battery includes voltage switching function such as 12/16/19Vand the maximum supply in 90 watts |
| Dimension | Length:150(mm)/width:100(mm) |

### 8.3. Multi-Media/Multi-Band Tested Results

Based on the experimental setup described in Section 8.2, the first test was done with pure water inside a water tank of 1 m in height and 1.8 m in width. More than 20,000 packets where received at distances of 1 m and 1.5 m. The second test was done with salty water inside the water tank; in this case, the tank was of 1 m in height and 8 m in width. Around 20,000 packets were received at distances of 4 m and 6 m. The final test was done in open seabed with highly turbid water. The seabed was constructed with a depth of 1 m depth and width of 12 m. Around 20,000 packets were collected distances of 2, 4, 6, 8, 10, and 12 m. In recent tests, the signal strength was collected only for VLC and IR.

Figure 14 shows the signal strength received by the multi-media/multi-band modem using pure water and the small water tank. The signal strength received for the VLC medium was 100 and 98.98 for 1 m and 1.5 m respectively, and that for the IR medium was 100 for both 1 m and 1.5 m.

Figure 15 shows the signal strength received using salty water and the large water tank. The signal strength received for the VLC medium was 29.2 and 80.1 for 4 m and 6 m respectively; that for the IR medium was not noted.

Figure 16 shows the signal strength received using the turbid water in open seabed test. In the case of 2 m, the signal strength received for the VLC medium was 100 for 1 m, 1.5 m, and 2 m, and that of IR is 100 for 1 m and 1.5 m. In the case of 4 m, the signal strength received for the VLC medium was 100, 100, 100, 95, 19, 0.07 for 1 m, 1.5 m, 2 m, 2.3 m, 3 m, and 3.8 m respectively, and for IR, 100 for 1 m and 1.5 m.

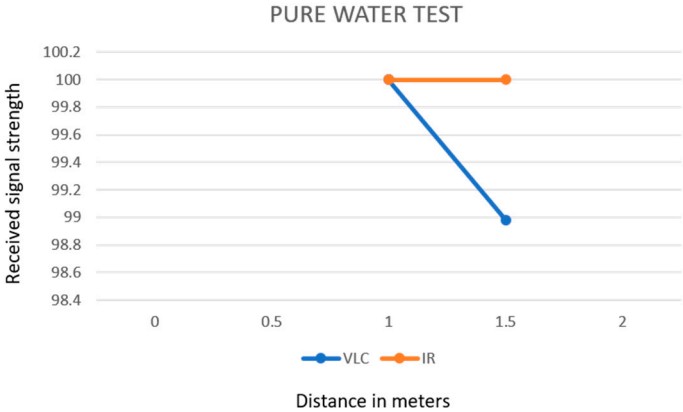

**Figure 14.** Pure water test using a small water tank.

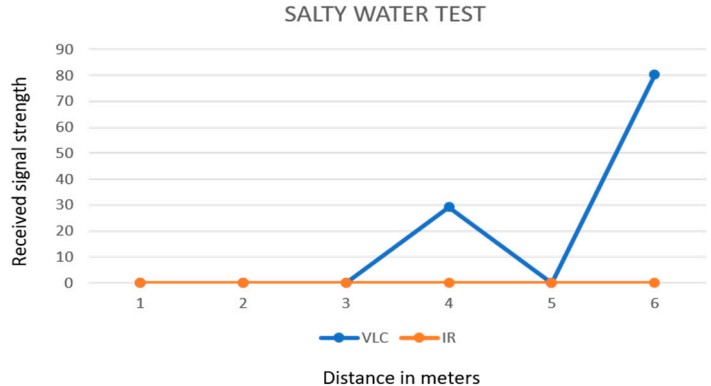

**Figure 15.** Salty water test using large water tank.

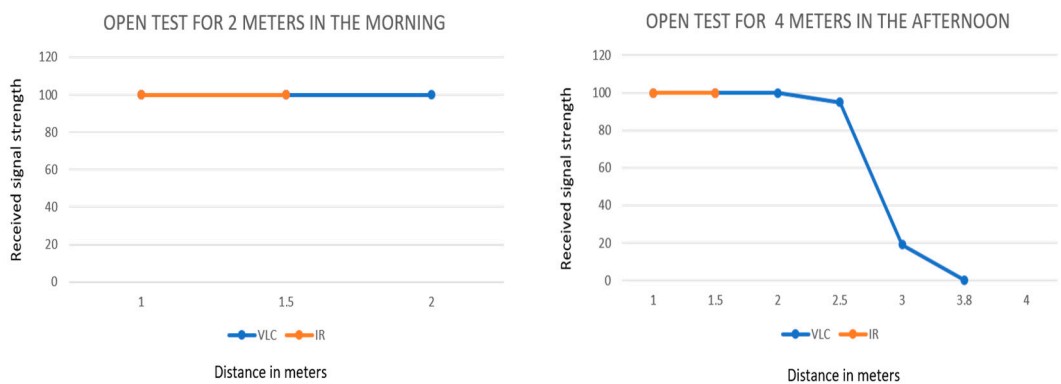

**Figure 16.** Open seabed test.

## 9. Conclusions and Future Work

In order to develop strong and reliable communication systems for underwater sensor networks, this paper proposes adaptation layer techniques using a multi-band/multi-media method inside the MAC layer. For multi-band communication, the existing communication protocol for MAC layers such as OFDM, OFDMA, or FDMA can be used to divide the frequency bands. The medium selection mechanism is proposed to carry the data through multiple media, such as acoustic, VLF, or IR. In the medium selection mechanism, the proposed scheme is split into two phases: (1) Finding the distance of near and far nodes; and (2) Medium selection and data transferring. To find the distance, the Manhattan method was used, and for the medium selection, a new algorithm was proposed. In the adaptation layered multi-media/multi-band approach, the RSSI between nodes is considered as the main factor to find the distance, since distance is the main property used when transferring data. Also, this paper takes into account scenarios such as diver networks monitoring and tsunami monitoring based on multi-media communication technology. In our approach, a multi-media/multi-band modem was developed with: (1) Two transducers supporting bandwidths of 70 kHz and 140 kHz for acoustic communication; (2) a single bandwidth for Infrared (IR) communication at a wavelength of 700 nm to 1 mm; and (3) Visible light (VL) communication that uses the blue wavelength at a range of 450 to 485 nm. So, this modem setup combines multiple bands for different media such as acoustic, VLC, and IR. Currently, tests have been undertaken only for VLC and IR. In future, red and green light will be considered for VL communication. Also, the multi-media/multi-band adaptation techniques will be improved by considering more properties of underwater environments, such as temperature, pressure, pH, etc.

**Author Contributions:** Conceptualization, S.-H.P. and S.-Y.S.; Supervision, S.-H.P.; Validation and Data curation, J.-I.N.; Visualization, S.-Y.S.; writing—original draft, D.R.K.M.; Writing—review & editing, S.-H.Y. and E.K.

**Funding:** This research is supported by Basic Science Research Program through the National Research Foundation of Korea (NRF) funded by the Ministry of Education (NRF-2019R1D1A1B03028903).

**Conflicts of Interest:** The authors declare no conflict of interest.

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
