# Peer review of "Multi-Media and Multi-Band Based Adaptation Layer Techniques for Underwater Sensor Networks"

_applsci, doi:10.3390/app9153187_

Round 1

Reviewer 1 Report

In this paper, the authors provide a comprehensive survey for underwater communications and networks. Some suggestions are presented as follows.

The authors are suggested to provide full names for some abbreviations in abstract.

Section 1.3 is suggested to discussed more with details.

Most figures are suggested to be reformatted in order to be presented clearly. 

Author Response

Point 1: The authors are suggested to provide full names for some abbreviations in abstract.

Response 1: The abbreviation terms are added inside the abstract such as AUV’s, OFDMA, FDMA, etc.

Point 2: Section 1.3 is suggested to discussed more with details.

Response 2: The points inside 1.3 is defined as below

1.Increase the lifetime of sensor nodes

2.Increase the reliability of data transmission

3.Improve the faster discovery of neighbour nodes

4.Reduce the transmission delay between nodes

5.Long stay connectivity between nodes

6.Faster medium selection mechanism to transfer data

As the points are shown as the benefits of multi-medium/multi-band communication technology in future. Base on the above points the medium selection mechanism is applied in section “7.4.2”. In this approach the fastest medium selection is satisfied based on distance as shown in the algorithm “7.4.2.2”.

The current development stage of multi-medium/multi-band communication is shown in section 8.1, 8.2 and 8.3. And the workflow of our modem is shown in “Figure 5”

Point 3: Most figures are suggested to be reformatted in order to be presented clearly.

Response 3: The figures are reformatted to make it better clarity and new figures are added inside the document such as Figure 12, Figure 13, Figure 14, etc.

Reviewer 2 Report

Please to see the attached file.

Author Response

Comments

Point 1: In Line 136, it’s written “..The speed of acoustic communication in underwater is 1500m/s.” It’s not an exact speed, as the author explains in the same paragraph. I think that a reference must be set, I suggest:

Response 1: As per the reviewer’s suggestion the speed of acoustic communication is changed to approximate value from 1480 m/s to 1540 m/s, based on the reference [2] and [3] as shown below.

·       Paul, C. (2003), Underwater acoustic modelling and simulation, 3 Ed, Spon Press of Taylor and Francis Group.

·       Barbeau, Michel & Garcia-Alfaro, Joaquin & Kranakis, Evangelos & Porretta, Steven. (2018). The Sound of Communication in Underwater Acoustic Sensor Networks. 10.1007/978-3-319-74439-1_2.

Point 2: In line 325, the Manhattan method is introduced, but no reference is set. I suggest citing some reference to complete the subsection

Response 2: The references for Manhattan method is added inside the text such as reference-number [58] and [59].

Point 3: The authors talk about two aspects of the modem: multi-band and multi-medium. The algorithm to choose the suitable technique medium-based is clear, but there are not comments about the frequency bands of operation. Some section is needed to clarify the operation of the modem in this aspect, because it is important to consider in a real case.

Response 3: In our approach the medium selection mechanism was clearly applied as shown in section 7.4. The consideration multi-band approach is explained in the section 8.2. Some of explanation are as follows.

·       In our approach the acoustic transduce supports bandwidth of 70 kHZ and 140 KHZ (Refer line number 538)

·       In IR communication only single bandwidth is used at the wavelength of 700 nm to 1 mm (Refer line number 539)

·       For Visible light communication blue light is considered now at the wavelength of 450 to 485 nm. Our future is to improve the multi-band approach using green and red light.

·       Workflow of our modem is shown in Figure 5-The frequency band splitter in Figure 5 will consider splitting of bandwidth for different medium such as acoustic, VLC or IR while switching. (Refer Figure 5)

Point 4: The most important section from my point of view, 7. Implementation setup and results, only takes less than 1 page and there is not any evidence of the setup done (e.g. a picture of the board working). I think is important to make some more tests to conclude the adaptation layer is suitable to operate in a real environment. In that sense, is the scenario of Figure 10 (line 382) real or simulated? The authors say nothing about the conditions taken into account. I think is needed before to give results of table 6 (line 389).

So, this Section (7) must be extended and be more convincing, for example, the board cited to implement the modem is Xilinx Zynq .. But, there are several boards, what model / version has been used? How much memory does the software takes? These questions are without any response in the current version of the manuscript.

Response 4: First, we are sorry to inform you that there was an error in “section 7”. During the first submission of paper, we entered estimated data based on the results of individual medium experiments conducted before this concept paper was presented. The co-authors commented that it was inappropriate to represent this concept because it was all done in a stable laboratory environment. Therefore, they wanted to show that the implementation of this concept has significant difficulty by presenting the real-world test value of the multi-media/multi-band modem they are currently developing. we sincerely apologize for the error made. Kindly neglect the “section 7” in previous version and consider the “section 8” in new version of document. Once again, we are extremely sorry for the situation. The current version of implementation setup done and gathered results are shown in section 8 as explained below.

                                                             Thank you.

Real implementation setup of our modem and its operations are shown in the section 8.

·       In subsection 8.1, multi-medium approach is tested using two medium such as acoustic and visible light. The specification of this modem is displayed in Table 5.

·       In subsection 8.2, multi-medium/multi-band approach is tested with different medium such as acoustic, VLC and IR with different bandwidth as explained in the lines (538, 539, 540). The specification of this modem is displayed in Table 6.

·       This approach can be suitable for the communication between divers in underwater network.

·       New figures are added, and some figures are updated inside our document such as Figure 12, 13, 14, 15 and 16.

Point 5: I miss in the manuscript having some results about power consumption, because as the authors say in the manuscript (line 146) “..The batteries in the constrained environment can’t be rechargeable”. So, is important to see how long the lifetime of the modem is when batteries are employed in a real situation. I suggest including some measures or estimations in that area, by means of figures / tables.

Response 5: The power consumption of tested multi-medium modem is shown in the specification shown in Table 5 and the power used for the tested multi-medium/multi-band modem is shown in Table 6.

Format defects

Point 1: The first reference in the manuscript is the [12] in line 135. All the references must be re-ordered in order of appearance, e.g. [12] à[1], and so on.

Response 1: All the references are rearranged in top-down format according to the reviewer’s suggestion.

Point 2: I suggest that tables 3, 4 and 5, are written in paragraphs more than as tables, because they take a lot of space and do not provide new research or contents. Perhaps, it is more suitable that these 3 sections would be located in a state-of-art Section before in the manuscript.

Response 2: Table 3, 4, 5 are reframed into paragraph as lot of spaces are occupied by the document.

Point 3: Some paragraphs have different line spacing, e.g. 228-241 is different that 242-253. The format must be kept in the whole document. Same occurs with the font employed, e.g. lines 321-322 have different font type than paragraph among lines 325-328. This is must be corrected.

Response 3: The different line spacing format is modified and made it common to whole document.

Point 4: Some paragraphs have different line spacing, e.g. 228-241 is different that 242-253. The format must be kept in the whole document. Same occurs with the font employed, e.g. lines 321-322 have different font type than paragraph among lines 325-328. This is must be corrected.

Response 4: The different line spacing format is changed and made it common to whole document.

Point 5: Formulae must be numbered in order of appearance following the specific format of the Journal (see the instructions for authors in the Journal’s web).

Response 5: The equation is changed as per the instruction given to author and the number is also applied in order.

Errata detected

Point 1: Table 1: The speed of signals of RF and Optical is wrong. I suppose is not “ x 108” but “ x 108 ” instead.

Response 1: The speed of RF and Optical communication value is changed from “2.3×108 ms−1” to “2.3×108 ms−1”.

Point 2: Points 1-7 in Section 4.1 should begin with capital letter, same way that the point 8 does.

Response 2: In section 4.1, the beginning of number list from 1 to 7 changed to capital letter.

Point 3: Section labelled as 4.1 in line 159, must be the Section 4.2 because “4.1” is already used in line 132. Same occurs:

·       With “4.1” in line 178, it should be the 5.1 Section.

·       With Section 5 (line 191), it should be the 6th Section (and the rest of sub-sections 5.x to 6.x)

·       With Section 6 (line 260), it should be the 7th.

·       (please to check the rest of Sections / Sub-sections)

Response 3: All the numbering mistakes appeared in the sections and subsections such as 4, 5, 6, etc. are corrected.

Point 4: In line 264, the manuscript says: “… So, Figure 2 shows the protocol stack…” , is not the Figure 2, is the Figure 3 instead. Same occurs in line 383, it must be written “Figure 10” and not “Figure 9”. Please to review the rest of numerations for the figures.

Response 4: The wrong appearance of figure number inside the text are corrected (for example Figure 2 to Figure 3) and the typos inside the documents are corrected.

Round 2

Reviewer 2 Report

Please to see the attached file. Thank you.

Author Response

Assessment: Accepted with minor corrections

Comments to the Authors

Reviewer comment number 1: Once again, the reference number does not fit the appropriate cite. In the references section, Paul (2003) is cited as [3] and Barbeau as [4]. However, in the manuscript (line 142) they both appears as [2][3] references. I think is an error, please to correct it.

Response 1: During our submission after the first revision “we missed to add the citation details in numbering 1 of references and was left blank”. This created mismatch in the reference numbering. Now it is corrected by moving reference numbers by one less, and the new reference number for the mentioned citations are “Paul (2003) cited as [2] and Barbeau as [3]”.

Reviewer comment number 2: In the line 437 of the revised manuscript, the text says: “…In [59] [60], the Manhattan method…” , so what are the correct references for Manhattan method? Are they [58][59] or [59][60]?. Please to check and correct it.

Response 2: During our submission after the first revision “we missed to add the citation details in numbering 1 of references and was left blank”. This created mismatch in the reference numbering. Now it is corrected by moving reference numbers by one less, and the new reference number for the mentioned citations are [59]and [60] for Manhattan method.

Reviewer comment number 3: In the lines 498-499, it is written: “Table 6 shows the RSSI value received as dB from the nearby node.” Obviously, it is referred to the Table 4 and not the 6. Apart of this, I warn you about the power consumption of 90W max in the Table 6: it is not suitable for underwater applications, because of a premature draining of the batteries. Such a high power is only understood in a lab experiment, so please to clear in the table that fact by means of a foot note or comment in the title of the table for avoiding confusion.

Response 2: “Table 6 shows the RSSI value….” Is changed to “Table 4 shows the RSSI value”.

The confusion in Table 6 is due to the misrepresentation of the word “electrical power source” as “Power consumption”. The same is modified in the table to avoid confusion.

·        The information entered in Table 6 is “electrical power source” used for our modem, with the voltage switching function and the maximum supply in watts.

·        The problem with multi-medium/multi-band communication technology is, how to operate multiple underwater communication modems at the same time; because the type of power received is different for different media.

·        The voltage required is 15 to 16 volts for acoustic medium, 12 volts for visible light medium and 5 volts for infrared medium.

·        In our approach we use the portable external battery with voltage switching function for different media in underwater. Now the switching function is operated manually and does not support automatic, this battery enabled us to validate the concept in underwater.

·        And starting from this year, our research team is trying to integrate the problem of receiving different power supplies for different media in the hardware.

Format defects and typos

Reviewer comment number 1: Reference [63] is never used in the text, only in the reference section. Please to use it or discard it.

Response 1: Reference number [63] is deleted, as we don’t have the reference number 63.

Reviewer comment number 2: Line 120: “…The speed of communication is 310^8 m/s. …”. I suppose the authors wanted to say “3 x 108 ” instead. The same error is located in Table 2, please to correct both of them.

Response 2: The speed of communication changed to “3 x 108 ” from 310^8 m/s” both in line number 120 and Table 2.

Reviewer comment number 2: Line 597: Reference 1 is empty! Be careful with reordering all the references due to this mismatch.

Response 3: During our submission after the first revision “we missed to add the citation details in numbering 1 of references and was left blank”. This created mismatch in the reference numbering. Now it is corrected by moving reference numbers by one less, and the new reference number is created.
